# Impact of Education for Sustainable Development on Cognition, Emotion, and Behavior in Protected Areas

**DOI:** 10.3390/ijerph19159769

**Published:** 2022-08-08

**Authors:** Yan Ding, Minyan Zhao, Zehong Li, Bing Xia, Zhanna Atutova, Dmitry Kobylkin

**Affiliations:** 1School of Geography and Ecotourism, Southwest Forestry University, Kunming 650224, China; 2Institute of Tibetan Plateau Research, Chinese Academy of Sciences, Beijing 100101, China; 3Institute of Geographic Sciences and Natural Resources Research, Chinese Academy of Sciences, Beijing 100101, China; 4Sochava Institute of Geography, Siberian Branch of the Russian Academy of Sciences, 664033 Irkutsk, Russia

**Keywords:** education for sustainable development, cognition, emotion, behavior, protected areas, structural equation modeling

## Abstract

Education for sustainable development (ESD) of protected areas is proposed to deal with global climate change and biodiversity conversation. It focuses on the “quality education” and “protection” of the United Nations’ sustainable development goals (UN SDGs), not only taking protected areas as the education place, but also as the theme and content of education. Based on cognitive-behavior theory and social emotional learning theory, this study constructs a “cognitive–emotion–behavior” dimension framework of ESD in protected areas, selecting Potatso National Park in Yunnan as a case study. Based on 529 valid visitor questionnaires, this study uses structural equation modeling to verify theoretical hypotheses, and analyzes the impact of ESD in protected areas on public cognition, emotion, and behavior. The results show that: (1) Cognitive and emotional factors jointly drive the behavioral intentions of ESD in protected areas, and social-emotional factors are slightly higher than cognitive factors; (2) Environmental knowledge, personal norms, nature connectedness, and places attachment positively affects behavioral intentions; (3) Indigenous knowledge has an impact on behavioral intentions through emotional mediation, and personal norms have an impact on behavioral intentions through direct effects; (4) Gender and visit frequency are important moderating variables in the ESD of protected areas. These conclusions provide the following suggestions for further development of ESD. First, by forming environment-friendly social norms and focusing on the mining and presentation of indigenous knowledge, the behavioral intention can also be enhanced to a certain extent; second, improving people’s emotion can also promote people’s behavioral intention, especially referring to optimizing nature connectedness, strengthening place attachment, and creating emotional connections; Third, specific groups of people should be taught specifically, and improve the supporting services of ESD.

## 1. Introduction

The implementation of Sustainable Development in protected areas (PAs) has become a necessary measure around the world [1,2,3]. In 2015, the United Nations formulated 17 goals for 2030, representing the world, is addressing a wide range of social and environmental problems in an interconnected global effort [4]. As an important component of Sustainable Development Goal 15 (SDG15), protected areas are not only the most effective measure for biodiversity conservation, but also play a key role in achieving other SDGs [5,6]. Protected areas are important natural classrooms of Education for Sustainable Development (ESD) [7,8]. Implementing ESD in PAs can also better exert environmental advantages and realize the educational function of PAs, as shown in Figure 1. On the platform of protected areas, ESD can reach full potential and become a cohesive and powerful engine.

Worldwide countries still face enormous challenges in achieving the SDGs, progress on most of the SDGs has been slow [9], and COVID-19 has even slowed down the pace of implementation [10], suggesting that much effort needs to be done to achieve the SDGs. There is an urgent need to find ways to accelerate their implementation. Seeking synergies among the SDGs is increasingly seen as a way forward for more effective implementation of the 2030 Agenda [11]. At present, PAs have been coupled with many SDGs to exert a strong synergistic effect [5]. Among them, target 4.7 ESD is the main means of promoting all other targets, and little research has been done between ESD and PAs. This research is based on Cognitive-Behavior Theory, Social Emotional Learning (SEL), and Situated Cognition, which integrates the advantages of SDG4 “Quality Education” and SDG15 “Life on Land”, and optimizes ESD in PAs according to the three dimensions of cognition, emotion, and behavior to construct the theoretical model, so as to contribute wisdom and strength to the realization of the 2030 sustainable development goals.

### 1.1. Theoretical Background

This research is based on Cognitive-Behavior Theory (CBT), integrating Social Emotional Learning (SEL) and Situational Cognition (SC), and designs ESD in PAs according to the three dimensions of cognition, emotion, and behavior, as well as constructing the theoretical model, as shown in Figure 2.

Cognitive-Behavior Theory holds that cognition, emotion, and behavior are inseparable, and one of them will continuously affect the other aspects [12]. It is people’s perception and cognition that shape emotion and behavior [13]; at the same time, emotions and behaviors also influence cognition process in continuous feedback [14,15]. The application of CBT to ESD in PAs will clarify the logical process of individual psychology and behavioral process, and help to improve the public’s cognitive level, positive emotions, and educational experience, and ultimately achieve the double balance of benign environmental behavior and environmental ecological protection. Social Emotional Learning is a new trend of curriculum learning in education research in recent years [16]. In view of the important role that social emotion plays in education, this paper selects social-emotion as the intermediary variable to replace the emotional variable in cognitive-behavior theory, it will help to strengthen the public’s experiential learning in nature, establish a deep connection with nature, cultivate the public positive attitude of respecting nature, and promote public’s action to protect nature. Brown, Collins, and Duguid systematically and completely expounded the theory of Situational Cognition in *Situation Cognition and Cultural Learning* published in 1989 [17], claiming that knowledge is linked to the specific context in which it is used, and cognition arises from real-time interaction with the environment [18]. The application of SC to ESD in PAs means that visitors learn, understand, and master knowledge in the context, and then achieve the internalization of knowledge and form their own unique cognitive cultivation.

In the process of advancing ESD, the three dimensions of cognition, social-emotion, and behavior are crucial [19]. The cognitive dimension includes the required knowledge and thinking skills, the socio-emotional dimension includes social skills, reflective skills, values, attitudes, and motivations, and the behavior dimension describes a variety of capacities [20]. Each SDG has specific learning goals in the cognitive, social-emotional, and behavioral aspects [21], so this study further refines ESD in PAs and condenses it into specific learning outcomes. Based on the above theoretical analysis, this paper defines and explains related variables such as environmental knowledge, nature connectedness, and place attachment, and puts forward related research hypotheses.

### 1.2. Research Hypothesis

#### 1.2.1. Cognition and Behavioral Intention

The popularization of knowledge is indispensable in ESD. In the context of PAs, visitors’ cognition is formed in three ways, namely environmental knowledge acquired in daily life, indigenous knowledge acquired locally, and personal norms embedded in perceptions.

The environmental knowledge involved in PAs mainly refers to the basic knowledge of natural ecosystem, which is the important factor that prompts individuals to carry out pro-environment behaviors. Environmental knowledge directly promotes environmental behavior [22], or indirectly drives behavior through environmental attitudes and behavioral intentions [23]. For a specific country, the relevance between environmental knowledge and environmental behavior might change with individual demographic characteristics [24]. Based on this, the H1 hypothesis is proposed:

**H1.** 
*Environmental knowledge has a significant positive effect on behavioral intentions.*


Indigenous knowledge is local knowledge possessed by local people or unique to a particular culture and society [25], which makes a valuable contribution to environmental behavior of the public. The socio-cultural aspects of indigenous knowledge can be used as a learning background for Pas [26]. Based on this, the H2 hypothesis is put forward:

**H2.** 
*Indigenous knowledge has a significant positive effect on behavioral intentions.*


Norms are the behavioral standards that we expect from ourselves or society, and they are the most direct factors among cognitive factors that drive individuals to engage in pro-environment behavior. Generally speaking, the higher an individual’s personal norms, the stronger his/her behavioral intentions. Based on this, the H3 hypothesis is proposed:

**H3.** 
*Personal norms have a significant positive effect on behavioral intentions.*


#### 1.2.2. Cognition and Social Emotion

When visitors visit PAs, the social emotions generated can be summarized into two types. Nature connectedness is the degree of personal integration with the nature and environment. Place attachment is the positive emotional relationship between an individual and a particular place, which is stronger than nature connectedness. The social emotions formed by people and places also affect individual attitudes and behaviors towards the environment.

Environmental knowledge is closely related to emotion. Nature connectedness is the emotional relationship between an individual and the environment. When individuals acquire a better understanding of the environment, they will bring about a deeper connection with nature [27]. Environmentally knowledgeable visitors will be more likely to appreciate, care, and show empathy for the environment [28]. From the perspective of mental health, contacting with nature improves mood via reducing the activity of the prefrontal cortex [29]. Based on this, the H4 and H7 hypotheses are proposed:

**H4.** 
*Environmental knowledge has a significant positive effect on nature connectedness.*


**H7.** 
*Environmental knowledge has a significant positive effect on place attachment.*


Indigenous knowledge makes an important contribution to environmental sustainability. Once indigenous knowledge is incorporated into environmental management, it can lead to stronger human-nature linkages [30]. For the public, the process of experiencing and participating in a particular region of knowledge can not only satisfy and stimulate their curiosity, but also increase knowledge, broaden horizons, and deepen place attachment. Based on this, the H5 and H8 hypotheses are proposed:

**H5.** 
*Indigenous knowledge has a significant positive impact on nature connectedness.*


**H8.** 
*Indigenous knowledge has a significant positive effect on place attachment.*


Personal norms are the most important driving factors of engagement in environmentally responsible behavior activities, and activate pro-social behavior in a direct manner [31]. Due to the vacancy of research on the impact of personal norms on environmental behavior through emotion, this study sought to explore the impact of personal norms on emotion. Based on this, the H6 and H9 hypothesis are proposed:

**H6.** 
*Personal norm has a significant positive effect on nature connectedness.*


**H9.** 
*Personal norm has a significant positive effect on place attachment.*


#### 1.2.3. Social Emotion and Behavioral Intention

Nature connectedness is an important predictor of environmental behavior, and the public who are closely connected with nature are more likely to participate in environmental behavior [32]. Nature-based education effectively improves ecological behavior by promoting nature connectedness and environmental knowledge [33]. Based on this, the following hypotheses are proposed:

**H10.** 
*Nature connectedness has a significant positive effect on behavioral intentions.*


**H12.** 
*Nature connectedness plays a positive and significant mediating role between environmental knowledge and behavioral intentions.*


**H13.** 
*Nature connectedness plays a positive and significant mediating role between indigenous knowledge and behavioral intentions.*


**H14.** 
*Nature connectedness plays a positive and significant mediating role between personal norms and behavioral intentions.*


Place attachment is considered as the starting point of environmental behavior, and its significant positive impact on environmental behavior has been confirmed [34]. However, the mediating role of place attachment from internal cognitive factors to external behavioral performance pathways needs to be further verified. Based on this, the following hypotheses are proposed:

**H11.** 
*Place attachment has a significant positive effect on behavioral intentions.*


**H15.** 
*Place attachment plays a positive and significant mediating role between environmental knowledge and behavioral intentions.*


**H16.** 
*Place attachment plays a positive and significant mediating role between indigenous knowledge and behavioral intentions.*


**H17.** 
*Place attachment plays a positive and significant mediating role between personal norms and behavioral intentions.*


Combined with the research hypotheses, a structural equation modeling of the impact of ESD in PAs on public cognition, emotion and behavioral intentions is constructed, as shown in Figure 3.

## 2. Materials and Methods

Based on the existing research, we make theoretical hypothesis, design questionnaires, and collect data around the variables involved in the relevant dimensions, and analyze the data collected by using structural equation modeling (SEM), the hypothesis of direct or indirect influence of all dimensions is verified to analyze the path of environmental behavior.

### 2.1. Participants and Procedure

This paper takes the Potatso National Park in Yunnan Province as a case study, which is actively building a demonstration base for ecological civilization education in China and shaping the image of a park education model. The Luorong Ecological School, Militang Ecological Experience Center and Bita Lake Ecological Education Library enhance the functions of education for sustainable development and enrich the recreation and educational products of national parks.

This study explores the relationship of public’s cognition, emotion, and behavioral intentions in Potatso National Park, and a questionnaire survey was conducted in Potatso National Park from April to October 2021. As it snows from November to March of the next year, the park is closed to visitors and is not allowed to enter. Questionnaires are issued by a combination of paper questionnaires and online questionnaires, and collected at the ecological education library and the waiting area of the park. After the pre-investigation analysis and questionnaire adjustment in the testing phase, the formal investigation phase was started, and the sampling quantity and quality were strictly controlled to ensure the representativeness and reliability of the samples.

The pre-investigation was conducted from 20 to 30 April 2021. Due to the impact of COVID-19 epidemic, the average daily number of visitors is about 1000. During this period, 60 questionnaires were distributed, and 49 questionnaires were valid, and field surveys also were conducted in the park. According to the analysis results of the test questionnaire, the items of the test questionnaire are adjusted and deleted to form the final formal questionnaire.

The formal survey was conducted from 1st May to the end of October, 2021. A total of 560 questionnaires were distributed, including 535 on-the-spot questionnaires collected by research assistants and volunteers, and 25 online questionnaires. Visitors filled out the electronic questionnaire via scanning the quick response code to complete the survey through Questionnaire Star that is a professional online questionnaire platform in China. The selection of the participants is by the convenience sampling method, and the participants cover a wide range of age groups. Because of random and incomplete answers, the 31 invalid questionnaires were excluded. The 529 valid questionnaires were actually obtained with an effective rate of 94.5%, which met the sample size required by the study.

### 2.2. Instrument

A questionnaire survey method was used to collect data about the constructs in the proposed model. The questionnaire was divided into two parts. The first part described the demographic information of participants, including age, gender, educational background, monthly income, activity mode, provinces, visit frequency, and volunteer identity.

The second part of the questionnaire contained a series of measurement items that were developed based on the three dimensions of cognition, social-emotion, and behavior. These items were adapted from other studies and slightly modified to be suitable for the context of PAs for ESD. These measurement items were rated on a five-point Likert scale from strongly disagree (1) to strongly agree (5). The research dimension and measurement items are shown in Table 1.

Cognition dimension divided three sub-dimensions with a total of 10 items including environmental knowledge, indigenous knowledge, and personal norms. Concerning the measurement items for environmental knowledge, we referred to the scales of Fremerey et al. (2014) [35] and Krasny et al. (2020) [36]. For the measurement items of indigenous knowledge, we referred to the scales of Geertz et al. (2000) [37], Berkes (2000) [26], and Wu et al. (2019) [38]. Regarding the measurement items for personal norms, we referred to the scales of Krasny et al. (2020) [39].

The social-emotion dimension divided two sub-dimensions with a total of six items including nature connectedness and place attachment. For the measurement items for nature connectedness, we referred to the scales of Krasny et al. (2020) [40]. The measurement items for place attachment used the scales of Daniel et al. (1992) [41], Jorgensen et al. (2001) [42], and Huang et al. (2006) [43].

The behavior dimension divided three sub-dimensions with a total of 10 items including engagement behavioral intentions, protected behavioral intentions, and environmental behavior. The measurement items for engagement behavioral intentions and protected behavioral intentions used the scales of Yang et al. (2019) [15]. For the measurement items for environmental behavior, we referred to the scales of Halpermy (2010) [44] and Luo et al. (2020) [45].

### 2.3. Analysis

In this study, SEM is used to test the theoretical model and research hypothesis. According to the research of Hair et al. [46], it is generally required that the ratio of observed variables to sample number of the model should be 1:10 to 1:15. There are 34 items in this research; in principle, the number of samples should not be less than 340 valid questionnaires. There are 529 valid samples in the study, which met the sample requirements for SEM, and AMOS was used as the SEM analysis software.

## 3. Results

### 3.1. Participant Characteristics

The collected 529 valid questionnaire data were entered into SPSS 24.0 software for analysis and processing, and participant characteristics analysis was conducted on the gender, age, educational background, monthly income, activity mode, generating provinces, visit frequency, and other aspects of the public in Potatso National Park, as shown in Table 2.

### 3.2. Reliability and Validity Test

In this study, Cronbach’s α analysis was carried out on the divided dimensions by confirmatory factor analysis. The Cronbach’s α coefficient of the total scale was 0.934, and all items were higher than 0.7, which have ideal reliability and stability, and meet the conditions for further verification of validity, as shown in Table 3.

As shown in Table 4, the combined reliability (CR) values of all dimensions in this study are greater than 0.6, indicating that the internal quality of the scale is excellent. At the same time, the average variance extraction (AVE) values of all dimensions are all above 0.5 [47], which indicates that the scale has good convergent validity and good internal quality of the scale.

### 3.3. SEM Model

Some indicators in the initial model fitting have not reached the standard, so the overall fitness of the model is re-corrected. The results of the overall fitting index of the revised model show that *χ*^2^ = 349.349, *df* = 121, *χ*^2^*/df* = 2.887, between 1–3, indicating that the model has a good parsimony; GFI = 0.926, AGFI = 0.896, NFI = 0.935, TLI = 0.944, and CFI = 0.956 are all greater than the generally accepted value of 0.90, RMR = 0.019 is less than the accepted critical value of 0.05, and RMSEA = 0.06 is less than the accepted critical value of 0.08, showing that the conceptual model of this study fits well [48].

The standardized path coefficient in the validation results shows that nature connectedness, place attachment, personal norms, environmental knowledge, and indigenous knowledge could directly or indirectly positively affect behavioral intentions. Among the 12 research hypotheses of direct effect, nine hypotheses of direct effect are valid, and three hypotheses of direct effect are not, as shown in Figure 4 and Table 5.

### 3.4. Mediating Effect

Among the six research hypotheses of the mediation effect, four mediation hypotheses are established, and two are not established, as shown in Table 6.

The direct effect of environmental knowledge on behavioral intention is significant, but the two mediating paths of environmental knowledge on behavioral intention are not significant, indicating that the mediating effect of nature connectedness and place attachment on behavioral intention through environmental knowledge is not significant.

The direct effect of indigenous knowledge on behavioral intention is not significant, and its impact on behavioral intention is entirely through the mediating effect. The mediating paths of indigenous knowledge on behavioral intentions are through nature connectedness and place attachment, respectively. Therefore, the mediating effect of nature connectedness and place attachment on indigenous knowledge and behavioral intention is a complete mediating effect, and the difference in mediating effect is not significant.

The direct effect of personal norms on behavior intention is significant. There are two mediating paths between personal norms and behavioral intentions, but they mainly influence behavioral intention through nature connectedness. Therefore, the mediating effect of nature connectedness and place attachment on personal norm affecting behavioral intention is partially mediating, and the difference in mediating effect is significant.

### 3.5. Moderating Effect

A multi-group analysis was used to test the moderating effect of gender, educational background, visit frequency, and volunteer experience on the relationships between indigenous knowledge, personal norms, nature connectedness, place attachment, and behavioral intentions. By comparing the fitting indexes of the constrained and unconstrained models, the differences in the models among groups were showed. There are two categories in volunteer experience groups, the samples with visit frequency were divided into only-once and more-than-once, educational background was divided into basic education and higher education, and gender was divided into two categories, male and female.

The model fit of these subgroups was tested. As demonstrated in Table 7, the value of metrics for each group was adequate. The *χ*^2^*/df* values of all the groups were less than 3, the CFI and NFI of the different groups were greater than 0.9, and the GFI of the different groups were close to 0.9, which are considered to be reasonable. The values of the RMSEA for all groups were mostly less than the maximum acceptable value of 0.05 [49]. Thus, the data for each group fit the model well, fulfilling the precondition for further analysis.

Next, comparing the unconstrained and constrained models. In order to measure the moderating effects of gender, educational background, visit frequency, and volunteer experience on the paths from latent independent variables (indigenous knowledge, personal norms, nature connectedness, and place attachment) to the latent dependent variable (behavioral intentions), the unconstrained models of the groups corresponding to the moderating variables were compared with their measurement weight models and structural weight models, respectively. If moderating effects exist, they should cause statistically significant differences in the path coefficient of the same two model variables within the subgroups. The chi-square difference test is used to examine the differences among these models. If the difference of the chi-square (Δ*χ*^2^) value between constrained and unconstrained models seems significant (*p* < 0.05), then it is clear that there is a difference between the different groups. However, if it is not significant, then those models are identical and characteristics of respondent do not work as a moderator in the mode.

The measurement weight model set the factors loading of the two subgroups to be equal [49]. In multi-group analysis, there are significant differences in the measurement weight model of the default hypothesis group. As shown in Table 7, the measurement weight models were not significantly different from the unconstrained models for the two subgroups corresponding to educational background (*p* = 0.244) and volunteer experience (*p* = 0.676), while the measurement weight models had significant differences with their unconstrained models in the subgroups corresponding to gender (*p* < 0.001) as well as visit frequency (*p* < 0.042). Thus, the invariance of the measurement weights model of the subgroups corresponding to educational background and volunteer experience were confirmed, meaning that the differences of the path coefficients among the latent variables of the structure model were not significant. However, the invariance of the measurement weights model of the subgroups of gender and visit frequency were unconfirmed, that is, the differences of the path coefficients among the latent variables of the structure model were significant. These results indicate that the effects of gender and visit frequency are significant as moderating variables while educational background and volunteer experience have insignificant moderating effects in the structure model.

The structural weight model added a setting based on the measurement weight model, which is the constraint that the regression coefficients between the latent variables are equal [49]. As the results demonstrated, the structural weight models were not significantly different from their unconstrained models for the three subgroups corresponding to gender (*p* = 0.001), educational background (*p* = 0.035) as well as visit frequency (*p* < 0.001), respectively. While the structural weight model had significant differences with the unconstrained model in the subgroups corresponding to volunteer experience (*p* = 0.313). Therefore, the invariance of the structural weights model of the subgroup corresponding to volunteer experience was confirmed, meaning that the differences of the path coefficients among the latent variables of the structure model were not significant. Meanwhile, the invariance of the structural weights model of the subgroups of gender and visit frequency were unconfirmed, that is, the differences of the path coefficients among the latent variables of the structure model were significant between the people who visit only once and those who visit more than once and also between males and females. However, the invariance of the structural weights model of the subgroups of educational background was unconfirmed, which was not the same as the result of the measurement weight model. This means that the effects of gender and visit frequency are significant as moderating variables while volunteer experience has insignificant moderating effects in the structure model, and the moderating effects of educational background need to be further verified.

#### 3.5.1. Gender Group

As Table 8 shows, personal norms (β = 0.274, *p* < 0.01), nature connectedness (β = 0.458, *p* < 0.001), and place attachment (β = 0.301, *p* < 0.01) on the behavior intention were all significant for males, and the direct effect of environmental knowledge (β = 0.185, *p* > 0.05) and indigenous knowledge (β = −0.202, *p* > 0.05) on the behavioral intention of males were not significant. For females, personal norms (β = 0.498, *p* < 0.001), nature connectedness (β = 0.259, *p* < 0.001), and place attachment (β = 0.277, *p* < 0.001), all significantly influenced their behavioral intention, while environmental knowledge (β = 0.083, *p* > 0.05) and indigenous knowledge (β = 0.131, *p* > 0.05) on the behavioral intention had no significant direct effects. The values of critical ratios (CR) for the differences between the parameters showed that the differences in value between males and females in the path from NC to BI was −1.518 (*p* > 0.05) and was −0.802 (*p* > 0.05) in the path from PA to BI. This indicated that these differences were not significant and, therefore, H4a and H5a are rejected. However, the difference in the path from PN to BI was 1.979, which indicated that these differences were significant and, consequently, H2a is accepted. The results of the path analysis are presented in Figure 5.

#### 3.5.2. Educational Background Group

As shown in Table 9, personal norms, nature connectedness, and place attachment on the behavior intention were not significant for basic education groups, while, for the higher education group, personal norms, nature connectedness, and place attachment, all significantly influenced their behavioral intention. For both groups, there were no significant and direct effects of their environmental knowledge and indigenous knowledge on their behavioral intention. In terms of the critical ratios for differences between parameters, the difference between the coefficients of each same path for the two groups was not significant. This result is consistent with the previous findings that the structural weight model for educational-background-groups was not significantly different from their unconstrained model, which means the rejection of H1b, H2b, H3b, H4b, and H5b. This result indicates that educational background has no moderating effects on behavioral intention between the basic-education group and higher-education group. The results of the path analysis are presented in Figure 6.

#### 3.5.3. Visit Frequency Group

From Table 10, it can be seen that the effects of personal norms (β = 0.313, *p* < 0.001), nature connectedness (β = 0.411, *p* < 0.001), and place attachment (β = 0.321, *p* < 0.001) on the behavior intention were all significant for only-once visitors, and the direct effect of environmental knowledge (β = 0.127, *p* > 0.05) and indigenous knowledge (β = −0.041, *p* > 0.05) on the behavioral intention was not significant. For more-than-once visitors, personal norms (β = 1, *p* < 0.001) significantly influenced their behavioral intention, while all nature connectedness (β = −0.053, *p* > 0.05), place attachment (β = −0.023, *p* > 0.05), environmental knowledge (β = −0.015, *p* > 0.05), and indigenous knowledge (β = 0.202, *p* > 0.05) on the behavioral intention had no significant direct effects. The values of critical ratios for the differences between the parameters showed that the difference in the path from PN to BI was 3.67, in the path from NC to BI was −3.377, and in the path from PA to BI was −2.548. This indicated that these differences were significant and, consequently, H3c, H4c, and H5c are accepted. The results of the path analysis are presented in Figure 7.

#### 3.5.4. Volunteer Experience Group

As shown in Table 11, both those with volunteer experience and without volunteer experience were significantly influenced by environmental knowledge, personal norms, nature connectedness, and place attachment, and neither group was directly influenced by their indigenous knowledge. Based on the values of critical ratios for differences between parameters, the difference in coefficient between environmental knowledge, personal norms, indigenous knowledge, and place attachment for the with volunteer experience and without volunteer experience groups was insignificant, while nature connectedness was significant. This result was not in accordance with the previous model comparison result, in which the structural weight model of the two volunteer experience groups did not differ from the unconstrained model. Therefore, the hypotheses of H1d, H2d, H3d, H4d, and H5d are rejected. The results of the path analysis are presented in Figure 8.

In conclusion, the moderating effects of gender and visit frequency were confirmed while the moderating effects of educational background and volunteer experience were not. This study indicated that gender has moderating effects on the relationship between personal norms and behavioral intention. Visit frequency moderated the relationship between personal norms and behavioral intention, nature connectedness, and behavioral intention as well as place attachment and behavioral intention.

## 4. Discussion

This research predicted environmental behavioral intention in the context of ESD in PAs based on the CBT. The results of this study demonstrate the partial utility of the CBT as a conceptual framework for predicting behavioral intention. The relationships between environmental knowledge, indigenous knowledge, personal norms, place attachment, nature connectedness, and the behavioral intention were examined. In addition, the moderating effects of gender, educational background, visit frequency and volunteer experience on these relationships were also tested to further explore the differences between different social demographic factors.

Whether environmental knowledge will lead to environmental behavior still needs to be discussed under different circumstances. This paper finds that environmental knowledge plays a significant positive impact on behavioral intentions, which is inconsistent with the conclusions of Liu [50] and Carmi [51]. A possible reason for this is that the public is in the specific situation of the PAs, which makes them easier to acquire environmental knowledge and promote environmental behavior. This is why we emphasize that environmental knowledge must be based on specific situations. In addition, the validity of this research hypothesis may be closely related to the achievements of the construction of ecological civilization in China, especially the increasingly perfect system of PAs, public awareness of the PAs is becoming clearer. Moreover, the important role of emotion in stimulating environmental behavior has been unanimously agreed by scholars. Only when the emotional system is activated and environmental emotions are aroused can the environmental behaviors be promoted. Therefore, in the specific circumstances of PAs, the role of environmental knowledge in activating the public emotional system can be focused on.

This finding incorporates indigenous knowledge into the cognitive dimensions of ESD in PAs in order to make up a deficiency of environmental knowledge. In this study, the direct effect of indigenous knowledge on the behavioral intention was not found to be significant, while the mediating effects of emotion were found to be significant. Furthermore, indigenous knowledge positively influences behavioral intention via nature connectedness and place attachment. There, although indigenous knowledge does not directly induce behavioral intentions, when combined with emotional situations, it can indirectly induce behavioral intentions. Thus, its role cannot be ignored. Indigenous knowledge offers different views on nature and science that generally differ from traditional education and offers rich and authentic contexts for science learning. This is consistent with the research the studies by Zidny et al. [52], that is, we should focus on research and practices of integrating indigenous knowledge with science education for sustainability. Indigenous knowledge and environmental knowledge should complement each other in visitors’ educational experiences. The introduction of indigenous knowledge in the PAs will represent specific cultural backgrounds and might help improve the interpretation of this knowledge, so that it makes knowledge more relevant to visitors. In addition, the incorporation of indigenous knowledge into ESD of PAs might help to enable visitors to gain positive emotional experiences and develop corresponding attitudes towards the environment.

The findings indicate that personal norms have a more significant direct effect on behavioral intention than environmental knowledge, nature connectedness, and place attachment. The influence of personal norms on behavioral intention was mainly produced through a direct effect, although the mediating effect of emotion also played a partial role.

This study selects the personal norms of the norm types to conduct structural equation modeling, and explores the influence of personal norms on environmental behavior intentions through emotion. However, just as Rebecca et al. [53] proposed that, whether the impact of norms on behavior intention varies according to the type of norms and behavior types, the ability to explain and predict environmental behavior intention by introducing personal norms in this research still needs to be further explored. Therefore, for other types of norms, such as descriptive norms and prohibitive norms proposed by Krasny [39], the ability to explain and predict the environmental behavior intention still needs additional research. In the content of ESD, it is necessary to emphasize the guiding and restraining role of social norms, establish a personal model of implementing environmental responsible behavior, and enhance the responsibility of individuals in environmental health. At the same time, the adverse consequences that may be caused by environmental problems, especially the impact on environmental health, should be publicized.

In addition, the relationship between personal norms and behavior intention was also found to be moderated by gender and visit frequency, but not by educational background or volunteer experience. Personal norms prominently influence the behavioral intention for only-once visitors and more-than-once visitors but was lower for only-once visitors. The possible reasons are that visitors who have visited more than once are so familiar with the PAs that the same knowledge and environment no longer enable them to generate new cognitions and emotions. At this time, personal norms become the main factor. Therefore, personal norms become the main influencing factor. In addition, it also shows that there is a solidification and non-innovation phenomenon in the ESD of PAs. For managers, they get ESD in PAs done once and for all and did not expand follow-up education resources for the ESD level in the PAs. In terms of the moderating effect of gender, the effect of personal norms on behavioral intention is stronger for females than males. Men tend to place more emphasis on accomplishing the outcome of a behavior, while women are more process oriented [49]. That is, men are more likely to be willing to put in more efforts to overcome constraints in order to maintain their personal norms when visiting PAs, while women placed more emphasis on using norms to restrain their behaviors at all times during the visit. Stern et al. [54] proposed that women have stronger beliefs than men about consequences for self, others, and the biosphere.

Nature connectedness and place attachment are also the direct significant factor in influencing the behavioral intention, which is consistent with previous research conclusions [40,44]. Emotion not only directly affects behavioral intention, but also indirectly affects behavioral intention as a mediating variable. Nature connectedness and place attachment captures the emotional component of human–nature interactions and influences environmental behaviors. The moderating effects of visit frequency on the relationship between nature connectedness and place attachment with the behavioral intention have confirmed that the effect of emotion on behavioral intention is greater for only-once visitors rather than more-than-once visitors. A possible reason for this is that the mysterious and strange visit tends to be followed with positive emotions. The first-time visitors can not only feel the freshness brought by PAs, but also feel the sense of belonging immersed in nature. There, the behavioral intention for only-once visitors is more influenced by their positive emotions than that for more-than-once visitors. The finding also suggests that the more positive emotion of visitors toward PAs is, the stronger behavioral intention is.

## 5. Conclusions

### 5.1. Findings

This study uses SEM to verify the functional relationship between cognition, emotion, and behavior variables in ESD, aiming to reveal the impact and role of ESD in PAs. The following findings are drawn:

(1) Cognitive and emotional factors jointly drive the behavior intentions of ESD in PAs. Emotion plays the greatest impact on behavioral intentions. The total impact of nature connectedness on behavior intentions (0.366) is higher than that of place attachment (0.275), indicating that the public’s nature connectedness is an important emotional factor affecting behavior intention. The deeper the public’s emotion, the more active their behavioral intentions. In addition, the influence of indigenous knowledge on behavioral intentions is mainly mediated by place attachment and nature connectedness. Cognition has the second highest impact on behavioral intentions, only lower than emotion. The total impact of personal norms on behavioral intentions (0.385) is higher than that of environmental knowledge (0.131), which reflects that cognition plays a great role in promoting behavioral intentions. It shows that improving the public’s cognition could stimulate their behavior intentions;

(2) The total impact of personal norms on behavior intentions (0.385) ranks first. It shows that personal norms are the most important influencing factor in cognitive factors. Personal norms play a leading role in behavior intentions, and mainly affect public behavior intentions through direct effects;

(3) Gender is an important moderating variable in ESD of PAs. The positive effect of individual norms on behavioral intention in the female group is significantly greater than that in the male group;

(4) Visit frequency is an important moderating variable in ESD of PAs. For the public who visited for the first time, nature connectedness and place attachment significantly and positively affected behavioral intention. For the public who had visited several times, the impact of personal norms on behavioral intention was greater than that for the public who visited for the first time.

### 5.2. Implication

In order to enhance the impact of ESD in PAs on public cognition, emotion, and behavior, this paper focuses on the three factors of personal norms, social emotion, indigenous knowledge, and moderating variables that affect the behavioral intentions of ESD in PAs, and puts forward the following three implications.

First, by forming environment-friendly social norms and focusing on the mining and presentation of indigenous knowledge, the behavioral intention can also be enhanced to a certain extent. In PAs, the forming of social norms is not only a warning and a reminder through interpretive signs, but all indigenous peoples and operators are the embodiment of norms and the foundation of environmental friendliness. Appropriate rewards and commend can be given to managers, staff, and local residents to shape the environmentally-friendly atmosphere within the PAs. In addition, by appreciating the public for completing an environmentally friendly walking tour, the social identity and self-identity of the visitors can be strengthened, and the behavioral intention of the public can be improved. ESD can be better achieved by internalizing social norms within the PAs into the personal norms of visitors.

Indigenous and local knowledge needs to be heard, and it is necessary to integrate these knowledge systems into ESD. The issue of “how they are told, who tells them, when they’re told” needs attention. Indigenous people are the main body of storytelling, and the public also prefer the traditional stories and customs told by local people. The interesting, educational, and entertaining indigenous knowledge is easier to meet the needs of the public to expand knowledge and deepen environmental behavior. In storytelling, conveying the emotion in the story is more important than just telling the knowledge. Only when the visitor is brought into the context can the visitor have a sense of substitution and immersion, and environmentally responsible behavior can be produced.

Second, improving people’s emotion can also promote people’s behavioral intention, especially referring to optimizing nature connectedness, strengthening place attachment, and creating emotional connections. Emotion improvement should focus on collective memory in daily life, which is not only the sum of individual memories, but also a public symbol objectified in social life. There is a need to both highlight the collective memory shared by the people of the PAs, including staff and local residents, and to make an impression on the collective memory of the visitor.

The protection of PAs is inseparable from the attention to society, culture, and people. For indigenous people, traditional festivals are commemorative events and are endowed with characteristic symbols. By learning about and participating in traditional festivals, visitors can gain knowledge in an emotional pleasure. At the same time, video ethnography can also be used to connect collective memory to create natural and cultural identity. For example, based on the principle of cultural respect, the local intangible cultural heritage is recorded through videos, and the natural environment and daily life of the indigenous peoples are recorded from different angles, so that the collective memory can be better continued. Through the integration of collective memory, a deeper nature connectedness and place attachment will be established, helping to realize ESD.

Providing visitors with on-site and follow-up collective memory points is important in enhancing emotion and environmental behavioral intentions. Through daily events such as “herdsman grazing, management and patrolling of parks, as well as animal and plant protection”, place attachment will be integrated into life situations. There is also an ongoing emotional bond after the visitor finishes their trip to PAs. The role of electronic equipment is very critical. Through Weibo, WeChat, and other public platforms, live broadcast platforms, the situation in the protected area is recorded in real time. In the continuous emotional cultivation, the visitor’s environmental behavior intention will also continue to improve.

Third, specific groups of people should be taught specifically and improve the supporting services of ESD. According to the characteristics of different types of visitors, creating ESD content is acceptable to “all the public and individuals”. It can not only meet the common needs, but also focus on the special characteristics. At the same time, the provision of knowledge and emotion needs to be constantly updated and improved.

## Figures and Tables

**Figure 1 ijerph-19-09769-f001:**
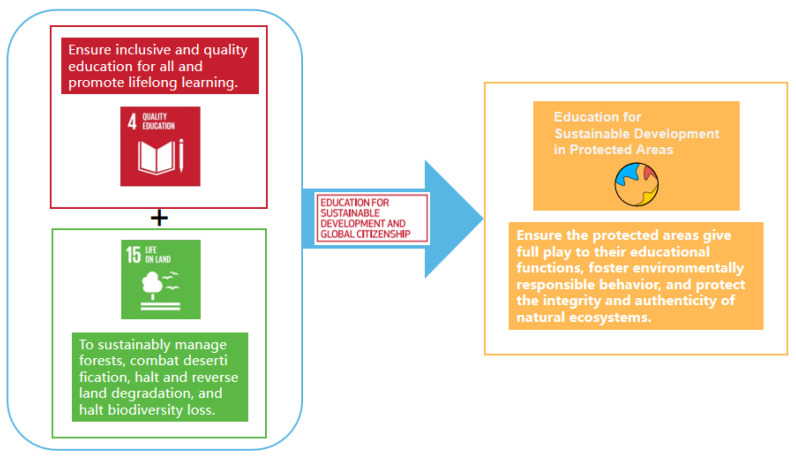
Theoretical diagram of education for sustainable development.

**Figure 2 ijerph-19-09769-f002:**
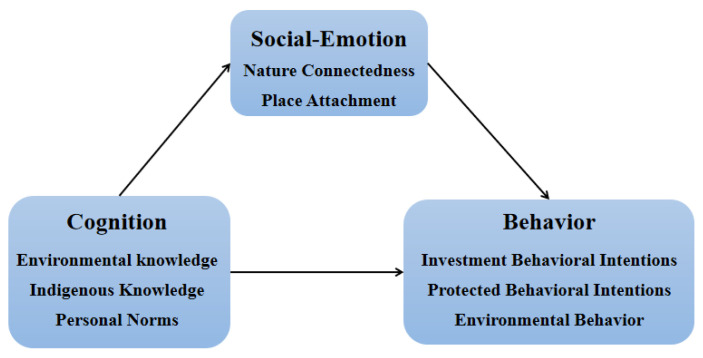
Theoretical model.

**Figure 3 ijerph-19-09769-f003:**
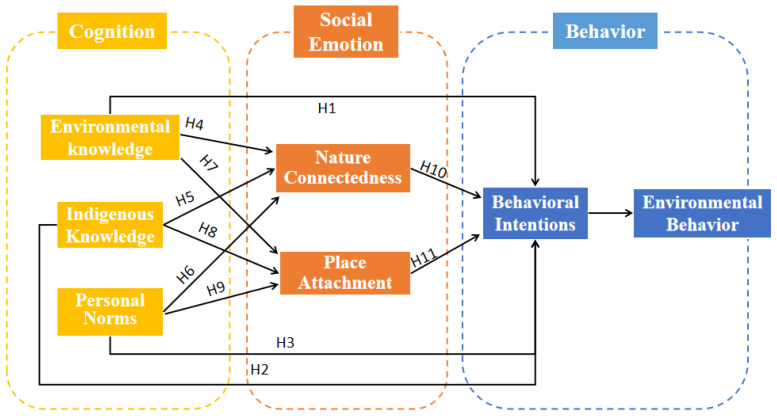
Hypothesized conceptual model.

**Figure 4 ijerph-19-09769-f004:**
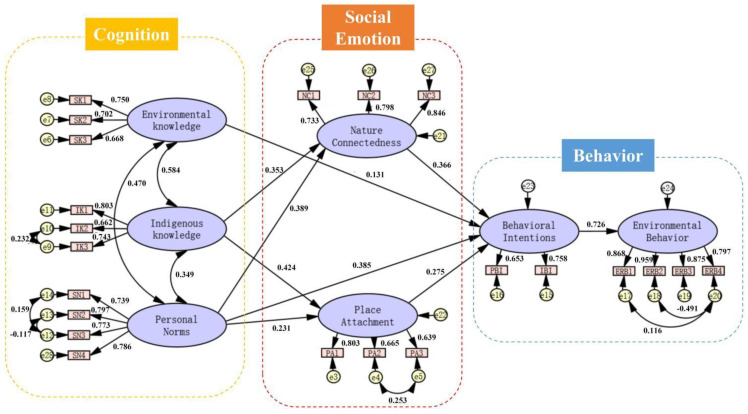
Theoretical model path coefficient.

**Figure 5 ijerph-19-09769-f005:**
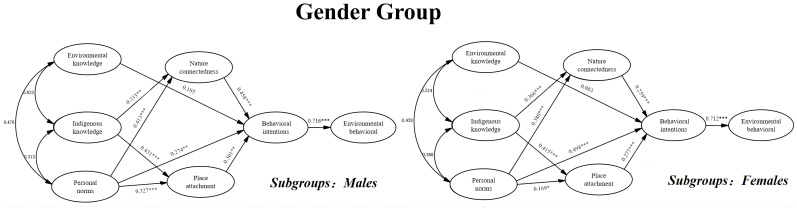
Path coefficient among gender group. Note: * *p* < 0.05, ** *p* < 0.01, *** *p* < 0.001.

**Figure 6 ijerph-19-09769-f006:**
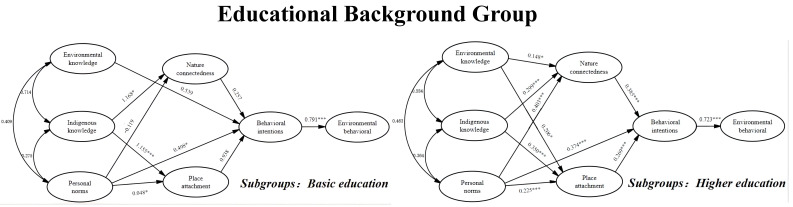
Path coefficient among educational background group. Note: * *p* < 0.05, ** *p* < 0.01, *** *p* < 0.001.

**Figure 7 ijerph-19-09769-f007:**
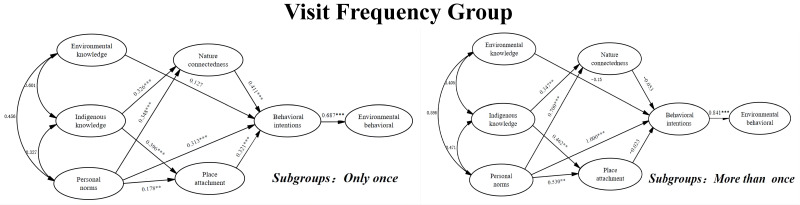
Path coefficient among the visit frequency group.

**Figure 8 ijerph-19-09769-f008:**
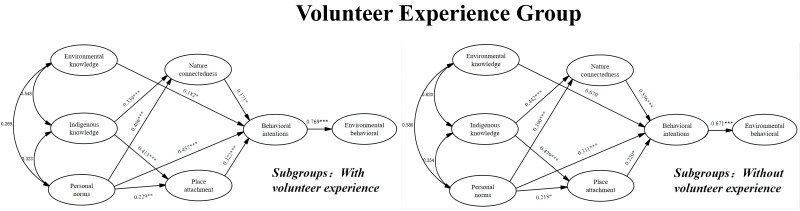
Path coefficient among the volunteer experience group. Note: * *p* < 0.05, ** *p* < 0.01, *** *p* < 0.001.

**Table 1 ijerph-19-09769-t001:** Visitor questionnaire.

Dimensions		Items	Sources
Cognition	Environmental knowledge	EK1: National Parks play an important role in the protection of natural ecosystems.	Fremerey, (2014) [35]; Krasny, (2020) [36]
		EK2: National parks are one of the most effective approach to conserve biodiversity.	
		EK3: National parks conservation could better response the challenge of climate change.	
	Indigenous knowledge	IK1: Potatso National Park is unique ecological environment and geological landscape.	Geertz et al. (2000) [37]; Berkes (2000) [26]; Wu et al. (2019) [38].
		IK2: Potatso National Park reflects the unique cultural traditions and regional customs of the Tibetan.	
		IK3: Potatso National Park is rich in biodiversity, with a wide variety of flora and fauna.	
	Personal norms	SN1: I am willing to abide by the rules and regulations in the park.	Krasny (2020) [39].
		SN2: If I break the rules, I feel guilty.	
		SN3: I think it is necessary to make rules and regulations.	
		SN4: We should be punished for violating social order and social morality.	
Emotion	Nature connectedness	NC1: I feel extremely relaxed and happy when walk in the nature.	Krasny, (2020) [40].
		NC2: Potatso is a great place to experience nature.	
		NC3: I love nature, so I want to protect it.	
	Place attachment	PA1: Potatso is special place where I could understand myself.	Daniel et al. (1992) [41], Jorgensen et al. (2001) [42]; Huang et al. (2006) [43].
		PA2: There is no other place to compare this place.	
		PA3: I hope I could stay here longer.	
Behavior	Engagement behavioral intentions	IBI1: I would like to share my journey of park with my family and friends.	Yang et al. (2019) [15]
		IBI2: I would like to visit Potatso National Park again.	
		IBI3: I am willing to publicize knowledge of environmental protection, animal and plant protection to others.	
	Protected behavioral intentions	PBI1: I am willing to be a volunteer to protect the environment.	
		PBI2: I am willing to contribute money and suggestions to protect the environment.	
		PBI3: I am willing to take environmentally friendly actions in the future.	
	Environmental behavior	EB1: I will pay attention to protecting the environment in my daily life.	Halpermy (2010) [44]; Luo et al. (2020) [45].
		EB2: I will abide by the tour rules of the park.	
		EB3: I will not destroy the environment, animals and plants of the park.	
		EB4: I will participate in and take actions that are beneficial to the environment.	

**Table 2 ijerph-19-09769-t002:** Participant characteristics.

Variables	Distribution	Frequency	Percent (%)	Variables	Distribution	Frequency	Percent (%)
Gender	Male	254	48	Tour mode	Personal	58	11
	Female	275	52		With Family	255	48.2
Age	Under 18	32	6		With Friends	187	35.3
	18–30	314	59.4		Group tour	18	3.4
	31–45	160	30.2		Other	11	2.1
	46–60	19	3.6	Monthly income	Less than 3000 Yuan	77	14.6
	Over 60	4	0.8		3000–5000 Yuan	81	15.3
Educational background	Middle school	20	3.8		5000–10,000 Yuan	176	33.3
	High school	31	5.9		1–1.5 million Yuan	76	14.4
	Bachelor’s degree	371	70.1		1.5–2 million Yuan	38	7.2
	Master’s degree and above	107	20.2		More than 2 million Yuan	81	15.3
Volunteer experience	Yes	256	48.4	Visit frequency	Only once	459	86.8
	No	273	51.6		More than once	70	13.2
Provinces	Yunnan Province	260	49.1				
	Other Provinces	269	50.9				

**Table 3 ijerph-19-09769-t003:** Reliability test.

Dimensions	Cronbach’s α	Items
Environmental knowledge	0.743	3
Indigenous knowledge	0.806	3
Personal norms	0.853	4
Nature connectedness	0.831	3
Place attachment	0.777	3
Engagement behavioral intentions	0.805	3
Protected behavioral intentions	0.801	3
Environmental behavior	0.920	4
Total	0.934	26

**Table 4 ijerph-19-09769-t004:** Validity test.

Dimensions	Items	Estimates	SMC	AVE	CR
Ideal value		>0.6	>0.36	>0.5	>0.6
Environmental knowledge	EK1	0.751	0.563	0.502	0.751
	EK2	0.709	0.502		
	EK3	0.663	0.440		
Indigenous knowledge	IK1	0.767	0.588	0.586	0.809
	IK2	0.726	0.527		
	IK3	0.803	0.644		
Personal norms	SN1	0.782	0.611	0.602	0.858
	SN2	0.767	0.588		
	SN3	0.785	0.616		
	SN4	0.77	0.593		
Nature connectedness	NC1	0.73	0.533	0.629	0.835
	NC2	0.797	0.636		
	NC3	0.848	0.719		
Place attachment	PA1	0.87	0.758	0.590	0.810
	PA2	0.634	0.402		
	PA3	0.782	0.611		
Engagement behavioral intentions	IBI1	0.738	0.545	0.559	0.791
	IBI2	0.704	0.496		
	IBI3	0.797	0.636		
Protected behavioral intentions	PBI1	0.701	0.492	0.567	0.796
	PBI2	0.701	0.491		
	PBI3	0.847	0.717		
Environmental behavior	EB4	0.876	0.767	0.776	0.933
	EB3	0.958	0.919		
	EB2	0.872	0.761		
	EB1	0.812	0.659		

**Table 5 ijerph-19-09769-t005:** Path coefficient and hypothesis test.

Research Hypothesis	Estimate	Std. Estimate	SE	t	*p*	Hypothesis Test
H1: EK→BI	0.14	0.131	0.062	2.238	*	True
H2: IK→BI	−0.021	−0.025	0.055	−0.391	0.696	False
H3: PN→BI	0.476	0.385	0.069	6.875	***	True
H4: EK→NC	0.103	0.092	0.075	1.376	0.169	False
H5: IK→NC	0.309	0.353	0.056	5.546	***	True
H6: EK→PA	0.183	0.13	0.106	1.724	0.085	False
H7: IK→PA	0.471	0.424	0.084	5.616	***	True
H8: PN→NC	0.5	0.389	0.07	7.167	***	True
H9: PN→PA	0.376	0.231	0.096	3.93	***	True
H10: NC→BI	0.352	0.366	0.055	6.434	***	True
H11: PA→BI	0.209	0.275	0.046	4.592	***	True
H12: BI→EB	0.658	0.726	0.048	13.608	***	True

Note: * *p* < 0.05, *** *p* < 0.001.

**Table 6 ijerph-19-09769-t006:** Mediating effect test.

Path	Environmental Knowledge to Behavioral Intentions				Indigenous Knowledge to Behavioral Intentions				Personal Norms to Behavioral Intentions				*p*	PM (%)	Hypothesis Testing
	Std-Estimates	SE	LLCI	ULCI	Std-Estimates	SE	LLCI	ULCI	Std-Estimates	SE	LLCI	ULCI			
H1: EK→BI	0.131	0.065	0.002	0.256									*	62.44%	
H13: EK→NC→BI	0.041	0.031	−0.013	0.109									0.127	18.55%	False
H16: EK→PA→BI	0.041	0.029	−0.01	0.106									0.1	18.55%	False
H2: IK→BI					−0.025	0.065	−0.168	0.092					0.585	−20.21%	
H14: IK→NC→BI					0.111	0.038	0.053	0.209					***	59.04%	True
H17: IK→PA→BI					0.114	0.039	0.052	0.207					***	60.64%	True
H3: PN→BI									0.385	0.072	0.26	0.543	***	65.94%	
H15: PN→NC→BI									0.144	0.037	0.086	0.236	***	24.04%	True
H18: PN→PA→BI									0.06	0.024	0.021	0.121	**	10.02%	True

Note: LLCI = lower limit confidence interval, ULCI = upper limit confidence interval. * *p* < 0.05, ** *p* < 0.01, *** *p* < 0.001.

**Table 7 ijerph-19-09769-t007:** The goodness-of-fit indices and results of comparison among the models.

Moderating Variables	Model	χ^2^	*df*	χ^2^/DF	Δχ^2^	ΔDF	*p*	TLI	CFI	GFI	ΔTLI	ΔCFI	ΔGFI	RMSEA
Gender	Unconstrained	810.871	377	2.151	-	-	-	0.919	0.934	0.878	-	-	-	0.047
	Measurement weight	852.539	392	2.175	41.668	15	0.000 ***	0.918	0.93	0.872	−0.001	−0.004	−0.006	0.047
	Structural weight	867.861	404	2.148	56.99	27	0.001 **	0.92	0.93	0.869	0.001	−0.004	−0.009	0.047
Educational background	Unconstrained	931.996	377	2.472	-	-	-	0.897	0.916	0.871	-	-	-	0.053
	Measurement weight	950.353	392	2.424	18.357	15	0.244	0.901	0.916	0.87	0.004	0	−0.001	0.052
	Structural weight	973.672	404	2.41	41.676	27	0.035 *	0.902	0.914	0.868	0.005	−0.002	−0.003	0.052
Visit frequency	Unconstrained	805.269	377	2.136	-	-	-	0.92	0.934	0.883	-	-	-	0.046
	Measurement weight	830.931	392	2.12	25.662	15	0.042 *	0.921	0.933	0.879	0.001	−0.001	−0.004	0.046
	Structural weight	867.55	404	2.147	62.281	27	0.000 ***	0.919	0.929	0.873	−0.001	−0.005	−0.01	0.047
Volunteer experience	Unconstrained	827.527	377	2.195	-	-	-	0.913	0.929	0.878	-	-	-	0.048
	Measurement weight	839.57	392	2.142	12.043	15	0.676	0.917	0.93	0.876	0.004	0.001	−0.002	0.047
	Structural weight	857.551	404	2.123	30.024	27	0.313	0.919	0.929	0.874	0.006	0	−0.004	0.046

Note: * *p* < 0.05, ** *p* < 0.01, *** *p* < 0.001.

**Table 8 ijerph-19-09769-t008:** Results of hypothesis for the moderating effects of gender.

Gender Paths	Male (*n* = 254)		Female (*n* = 275)		CR	Result
	Effect	*p*	Effect	*p*		
H1a: EK→BI	0.185	0.076	0.083	0.208	−0.725	Rejected
H2a: IK→BI	−0.202	0.083	0.131	0.068	2.541	Rejected
H3a: PN→BI	0.274	**	0.498	***	1.979 *	Accepted
H4a: NC→BI	0.458	***	0.259	***	−1.518	Rejected
H5a: PA→BI	0.301	**	0.277	***	−0.802	Rejected

Note: * *p* < 0.05, ** *p* < 0.01, *** *p* < 0.001.

**Table 9 ijerph-19-09769-t009:** Results of hypothesis for the moderating effects of educational background.

Educational Background Paths	Basic Education (*n* = 51)		Higher Education (*n* = 477)		CR	Result
	Effect	*p*	Effect	*p*		
H1b: EK→BI	0.539	0.504	0.11	0.073	−0.507	Rejected
H2b: IK→BI	−0.907	0.618	−0.012	0.85	0.493	Rejected
H3b: PN→BI	0.498	0.049	0.374	***	−0.142	Rejected
H4b: NC→BI	0.257	0.654	0.385	***	−0.001	Rejected
H5b: PA→BI	0.938	0.367	0.269	***	−0.469	Rejected

Note: *** *p* < 0.001.

**Table 10 ijerph-19-09769-t010:** Results of hypothesis for the moderating effects of visit frequency.

Visit Frequency Paths	Only Once (*n* = 459)		More than Once (*n* = 70)		CR	Result
	Effect	*p*	Effect	*p*		
H1c: EK→BI	0.127	0.06	−0.015	0.876	−1.278	Rejected
H2c: IK→BI	−0.041	0.564	0.202	0.074	1.893	Rejected
H3c: PN→BI	0.313	***	1	***	3.67	Accepted
H4c: NC→BI	0.411	***	−0.053	0.665	−3.377	Accepted
H5c: PA→BI	0.321	***	−0.023	0.827	−2.548	Accepted

Note: *** *p* < 0.001.

**Table 11 ijerph-19-09769-t011:** Results of the hypothesis for the moderating effects of volunteer experience.

Gender Paths	With (*n* = 256)		Without (*n* = 273)		CR	Result
	Effect	*p*	Effect	*p*		
H1d: EK→BI	0.182	*	0.07	*	−0.95	Rejected
H2d: IK→BI	−0.041	0.66	0	0.997	0.29	Rejected
H3d: PN→BI	0.457	***	0.311	***	−1.23	Rejected
H4d: NC→BI	0.171	*	0.556	***	3.363	Rejected
H5d: PA→BI	0.322	***	0.22	**	−0.803	Rejected

Note: * *p* < 0.05, ** *p* < 0.01, *** *p* < 0.001.

## Data Availability

Some or all data and models that support the findings of this study are available from the corresponding author upon reasonable request.

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
