# Peer review of "Impact of Education for Sustainable Development on Cognition, Emotion, and Behavior in Protected Areas"

_ijerph, 2022, doi:10.3390/ijerph19159769_

Round 1
Reviewer 1 Report
The authors are to be acknowledged for exploring the corresponding SDGs-related cognitive, emotional and behavioural indicators of citizens within a protected area in China. Their research design and performance tend to fall within the significant UN SDGs exploration combined with interesting conceptual and structural equation modelling. Therefore, their study seems to be of considerable merit to the International Journal of Environmental Research and Public Health. If the authors kindly agree, I think that some minor editing is needed to support the manuscript's final improvement. In particular, the authors might find useful the following detailed points, as next.
Lines 46-47 on p. 2: the words Figure 1 and the title of the diagram Theoretical diagram of the education for sustainable development are reported 2 times. Please kindly remove double wording. Line 68 on p. 2: the word Figure and Theoretical Model are mentioned 2 times. Please kindly remove double wording. Line 110 on p. 3: specific instead of specifific, Line 115 on p. 3: people instead of peoples. Line 134 on p.4 “the” environment line 144 on p. 4 Indigenous with small i. Line 186 on p. 5 Figure and Hypothesized conceptual model are reported 2 times. Line 237 on p. 6: the word Table is mentioned twice. Visitor Questionnaire can be inserted in brackets. Line 269 on p. 8 the words Table and Participant Characteristics are mentioned twice. Line 276 on p. 8: the words Table and Reliability Test are reported twice. Line 278 on p. 9: the word Table is mentioned twice. Line 294 on p.10 Figure, Table and Theoretical Model Path Coefficient and Path Coefficient and Hypothesis Test are mentioned twice. Line 304 on p. 11 the words Table and mediating effect test are reported twice. Line 334 on p. 12 the word Table is mentioned twice. Line 356 on p.12 the word Table is mentioned twice. Line 396 on p. 14 the word Table is reported twice. Line 410 on p. 14 the word Figure is mentioned twice. Line 416 on p. 14 the word Table is reported twice. Line 419 on p. 15 groups instead of group. Line 428 on p. 15 the word Figure is mentioned twice. Line 434 on p. 15 the word Table is mentioned twice. Line 446 on p. 16 the word Figure is reported twice. Line 452 on p. 16 the word Table is mentioned twice. Line 463 on p. 16 the word Figure is reported twice. Line 475 on p. 17 the words in specific, 1) can be added to ease sentence connectedness. Line 478 on p. 17 the environmental knowledge plays. Line 479-481 could you please rephrase the sentence for better comprehension? Line 482 on p. 17 validity/merit instead of valid. Line 487-488 can the environmental behaviors be promoted Line 493 on p. 17 Rebecca et al [41] Line 498 on p. 17, Krasny might need a reference number in brackets added here. Line 562 on p. 18 Indigenous might be written in small i, people instead of peoples.
Reviewer 2 Report
The paper undertakes a great effort to understand how environmental awareness is formed. However, there are also a few shortcomings:
1. The presented statistics, e.g. Cronbach's alpha, are not suited for questionnaire data. Other measures, like Ordinal Alpha, can solve this problem, which ells can cause both false positives and negatives. Se Liddell
2. The aspect of education, clearly underlined in the introduction seems to disappear before the discussion. Something like "how the protected areas can give full play to their educational functions" (figure 1) should be integrated into a discussion of the topic.
3. The educational aspect seems to be knowledge distribution solely. An additional discussion could be if skills and competencies could be part of these educational functions.
In total, the hypothesises are interesting, and the dataset seems well-collected, but the processing and discussion leave a little behind
Analyzing ordinal data with metric models: What could possibly go wrong? Torrin M.Liddell and John K.Kruschke
Reviewer 3 Report
The paper is well written, and is quite interesting. The paper provides us with the important methodology for analyzing education for sustainable growth. The method is technically sophisticated, and derives interesting results.
How the cognitive and emotional factors improve the education for sustainable development (ESD) is the main question. The interesting point in question is the relation between emotion and effectiveness in education. This part is quite original point in the field.
The field of ESD is attracting attention in the world. The paper introduces the Cognitive-Behavior Theory (CBT), and this expand the horizon of ESD. References related with Cognitive-Behavior Theory are included in the reference list, and this list is quite relevant for this new area.
The paper is distinguished because CBT is applied to ESD with utilizing structural estimation model, and clarifies the strength of effect of cognition on emotion, and behavior. It is quite interesting that the hypotheses are tested using SEM.
The discussion based on the empirical results should be extended to the curriculum and learning contents. Especially, how positive emotional factors would be stimulated by what type of contents would be important.
Finally, it would be better if theoretical explanation on the difference in the effect among groups are added to the conclusion.
Minor comment:
Line 68: Figure Figure 2 should be Figure 2.
Round 2
Reviewer 2 Report
Thanks for the detailed reply.
While I would not personally apply these statistics to this data, I believe the presented arguments are valid, at least for comparison with existing literature. Ordinal alpha might not be the right choice either, but I can see that you understand the caution needed in cases like this. Maybe a tiny part of the arguments you presented here should go into the study's limitations.
The updated discussion improved my understanding of the paper.